# Impact of Inclusive Leadership on Innovative Work Behavior: The Mediating Role of Job Crafting

**Yinping Guo, Junge Jin and Sang-Hyuk Yim \***

School of Business Administration, Dankook University, Yongin 16890, Republic of Korea
\* Correspondence: shyim310@dankook.ac.kr; Tel.: +86-031-8005-3416

**Abstract:** The study aims to examine the mediating role of job crafting between inclusive leadership and innovative work behavior. The data were collected from 314 workers employed in China's small and medium-sized industries. The data collection was done through survey design. The data analysis was done using Spss 26.0 and through structural equation modeling by Mplus 8. Inclusive leadership was found to be related to job crafting and innovative work behavior of the employees. Job crafting was found to be mediating between inclusive leadership and innovative work behavior. The study delineated the link mechanism between inclusive leadership and innovative work behavior. Studying inclusive leadership in the context of Chinese culture is a powerful complement to inclusive leadership theory. This paper provides the managers of SMEs with significant managerial insights into how inclusive leadership can effectively motivate employees' innovative work behaviors.

**Keywords:** inclusive leadership; job crafting; innovative work behavior; belongingness; uniqueness

## 1. Introduction

In the post-modern era, markets have become more fluid, flexible and changing, and the natural way organizations and institutions respond to this change is by making themselves fluid, flexible and changing (Brinkmann 2010). The present profoundly aggressive and technology-driven businesses s push the importance of knowledge workers to a new height (O'Donovan 2020). Innovation has become the hallmark of the long-term achievement and survival of small and medium-sized enterprises (SMEs) and the crucial source of creating competitive advantages for all businesses over their rivals. (Smith and Tushman 2005). Due to the short product life cycle and globalization, no firms could grow and develop well without innovation (Lenka and Gupta 2019).

Unlike in the past when the search for innovation was restricted to a small quantity of intelligent employees (Ziemnowicz 1942), the current pursuit of innovation is expected and should be done by all the employees of the organization. Innovative work behavior (IWB) is critical not only for the high-tech industry but also for the entire organizational workforce (De Jong and Den Hartog 2010). With their acumen for coming up with new ideas, they often play a key role in the innovation and development of companies, thus becoming a vital resource for enterprises (Hirst et al. 2009). Businesses are showing a growing interest in how to motivate employees to come up with innovative ideas and put them into practice to improve performance. Academic researchers have also begun to find new ways to promote IWB among employees (Khan et al. 2020). Thus, a growing body of literature has focused on exploring the factors that motivate and facilitate IWB among employees (Scott and Bruce 1994), especially in high-tech SMEs (Bagheri 2017).

Leadership is considered to be a particularly important factor affecting the creativity and innovation of members within an organization (Mumford and Hunter 2005). The centrality of knowledge workers calls for a fundamental change in leadership, as the appropriate leadership style has the potential to help employees pursue IWB (Watts et al. 2020). As employees move from an era of productivity to an era of innovation (Lenka

and Gupta 2019), there is a need to change the way leaders engage with them. As the hallmark of the knowledge-based economy, innovation requires a completely different leadership philosophy which gives employees the freedom to be creative and supports them to innovate (Uhl-Bien et al. 2007). LMX theory argues that with respect to differences in relationships between in-group members and out-group members and leaders (Gerstner and Day 1997), leader-follower relationships have been found to be essential for a variety of work outcomes.

Inclusion is considered as a unique diversity management approach (Roberson 2006), which focuses on satisfying employees' needs for uniqueness and belongingness at the same time (Shore et al. 2011) so that employees can become their unique selves at work and experience the feeling of "home" simultaneously, and actively solve the problem of diversification (Shore et al. 2009). As a result, employees can unleash their full potential, tackle challenges, collaborate with others, and improve their work experience (Shore et al. 2011) and thus be prone to be motivated to engage in IWB. Inclusive leadership is one of the contextual factors contributing to inclusion, alongside inclusive climate and inclusive practices (Shore et al. 2011). Inclusive leaders here refer to those who demonstrate openness, accessibility, and availability in their interactions with followers (Nembhard and Edmondson 2006). Tierney has pointed out that although there is a growing body of empirical research on the impact of leadership on creativity, so far this research is still in its infancy (Tierney 2008). At present, most studies on inclusive leadership were carried out in Western countries. Different from the individualistic culture of western countries, China is a country with a typical collectivist culture, and China has advocated an inclusive spirit since ancient times. Therefore, the study of inclusive leadership in the Chinese cultural background is a favorable supplement to this theory. In China, the study of inclusive leadership is still in the development stage. In addition, through the literature review, we know that most studies on the mechanism of inclusive leadership and innovative behavior focus on psychological aspects, such as psychological safety (Javed et al. 2019b), psychological empowerment (Javed et al. 2019a), self-efficacy (Qi et al. 2019), etc. Different from previous studies, our study puts more emphasis on employees' initiative behaviors. As IWB is an unconventional behavior, which usually avoids traditional methods and explores and implements new working methods when approaching work, employees need a special form of active working behavior, including employees' initiative to change the (perceived) characteristics of the work (Tims and Bakker 2010), where job crafting acts as this behavior. Recently, Javed et al. called for research on other mediating variables to specifically examine the mechanism of the relationship between inclusive leadership and IWB (Javed et al. 2018). In response to this call, this study casts job crafting as an extra-role behavior in the relationship between inclusive leadership and IWB.

Unlike other forms of Job design which are usually seen as a top-down process in which the organization creates jobs and managers supervise and order people for achieving the pre-designed job tasks, job crafting is a proactive work behavior that involves changing and reshaping the tasks or relationships to keep the job challenging, motivating and healthy (Kim et al. 2018). The causal links between inclusive leadership and IWB are through the employee-initiated changes in the job -job crafting (Wrzesniewski and Dutton 2001)–that in turn is influenced by inclusive leaders facilitating group members feeling part of the group and retaining their sense of individuality. According to the LMX theory, we propose that inclusive leadership through its acknowledging employees' input and facilitating their work can instill leadership trust in employees who in turn will feel confident to bring changes into their jobs, thus not only enhancing their job resources but also making the job more challenging for them. With these additional resources and positive emotions, employees will find they are motivated to create new ideas and implement them.

The study contributes to the existing knowledge base in multiple ways. First, the study tests the linking mechanism between inclusive leadership and IWB. This is the first study that uses job crafting as a mediator between inclusive leadership and IWB. Additionally, we studied job crafting through its behavioral dimensions separately instead of seeing

it as a high-ordered construct to assess in detail the influence of each of the different dimensions, and enrich the understanding of the job crafting theory. Finally, we tested our model in China, a country with a long history of an inclusive spirit where the inclusive culture will deepen our understanding and provide a different angle to see this inclusive leadership theory.

## 2. Literature Review and Hypotheses Development

### 2.1. Inclusive Leadership

In diverse work groups, meeting employees' needs for uniqueness and belongingness is primarily the role of leaders (Nishii and Mayer 2009). The concept of inclusive leadership was first proposed by Nembhard & Edmondson in 2006 (p. 927) as "words and deeds by a leader or leaders that indicate an invitation and appreciation for others' contributions" and then further developed into "leaders who exhibit visibility, accessibility, and availability in their interactions with followers (Atwater and Carmeli 2009). More recently, it is defined as a set of leader behaviors with a focus on facilitating belongingness by ensuring justice and equity and providing shared decision-making opportunities, while valuing uniqueness by encouraging diverse contributions and helping group members fully contribute (Randel et al. 2018). Compared with other leadership styles, inclusive leadership places more emphasis on individuals, both supporting and facilitating belongingness and indicating a value for uniqueness to promote their diverse contributions and abilities.

By creating a comfortable environment and expressing support for followers and their opinions, inclusive leadership makes members feel comfortable and conveys that they have the members' best interests in mind (Nembhard and Edmondson 2006) so as to let the follower feel supported (Bannay et al. 2020). Inclusive leadership not only indicates respect for individuals but also creates a system to ensure justice and equity (Randel et al. 2018). Ensuring decision-making control is distributed over specific aspects of the work and creating decision-making sharing practices are used by inclusive leadership in ensuring shared decision-making (Randel et al. 2018).

Inclusive leaders encourage diverse contributions by paying special attention to soliciting different perspectives and approaches and creating an environment that welcomes diversity (Winters 2013). By trying to understand their members' strengths and preferences, and convincing them to bring their full selves to their work, inclusive leaders can help team members contribute fully (Randel et al. 2018).

### 2.2. Job Crafting

Job crafting is a proactive work behavior, which is employee-initiated changes in the task, relation and cognitive boundaries of work (Wrzesniewski and Dutton 2001). To address the difficulty of developing a single scale to measure job crafting across industries, the job crafting definition was proposed from the perspective of job demands and resources using the job demand and resource model (JD-R) (Tims and Bakker 2010). According to the JD-R model, Job demands are aspects of a job that require sustained physical or mental effort and are therefore associated with certain physical and psychological costs (Demerouti et al. 2001), which could be used by employees when they craft their jobs by increasing challenging job demand and decreasing hindering job demand (Demerouti et al. 2001). Job resources are aspects of a job that help enhance job performance by increasing structural resources and/or social resources when employees craft their jobs (Tims and Bakker 2010).

Compared with the traditional job enrichment approaches which are imposed from the top to motivate employees, employee-initiated job crafting is more beneficial for employees and the organization (Hakanen et al. 2017). Seeking challenges, reducing hindering demands and expending resources promote employees' well-being (Hakanen et al. 2017), satisfaction (De Beer et al. 2016) and person-job fit (Kooij et al. 2017).

Tims and Bakker (2010) suggest four dimensions of job crafting: (1) increasing structural job resources, (2) increasing social job resources, (3) increasing challenging job demands, and (4) decreasing hindering job demands. Job crafting in the form of increasing job

resources and increasing job challenges (but not reducing hindering job demands) has been found to be positively related to both well-being (Tims et al. 2013)), and performance (Tims et al. 2012). Different results appear when specific job crafting dimensions are considered. For example, increased challenging job demands were related to other assessed job performance, while reduced hindering job demands were associated with turnover intention (Rudolph et al. 2017). Therefore, we believe that job crafting dimensions with exception of reducing hindering work demands are closely related to our research. In addition, the results of the meta-analysis relative weight analysis illustrate the unique relationships between all four job crafting dimensions and different job outcomes (Rudolph et al. 2017). In addition, the results of the meta-analysis relative weight analysis illustrate the unique relationships between all four job crafting dimensions and different job outcomes. Therefore, in our study, we will examine the mediating effect of job crafting through individual dimensions.

### 2.3. Innovative Work Behavior

IWB is defined as the act of intentionally creating, introducing, and applying new ideas in a work role, team, or organization to benefit role performance, team, or organization (Janssen 2000). According to Scott and Bruce (1994), IWB is complex and multistage work behavior including idea generation, idea promotion, and idea implementation, which are not required to be in strict sequence, employees can engage in any combination of these behaviors at any time, thus complicating the entire IWB process (De Jong and Den Hartog 2007). IWB is an extra-role behavior that requires employees to reject stereotypes, go beyond the existing conventions, find new technologies, develop new ways to achieve organizational goals, and apply new working methods (De Jong and Den Hartog 2010). Despite the reality that innovation and creation are often used interchangeably, innovation differs from creativity in that it emphasizes not only the ability to generate new ideas but also their implementation (Scott and Bruce 1994). Therefore, the generation of new ideas can only be a part of innovation (Qi et al. 2019). In addition, innovation does not emphasize doing things for the first time, which makes the adoption of outsider products and processes part of innovation (Scott and Bruce 1994). IWB is found to positively affect the innovative performance of the firm (Sanz-Valle and Jiménez-Jiménez 2018). Employees who manifest IWB have a high satisfaction level (West and Anderson 1996) and better performance (Leong and Rasli 2014), which therefore leads to the promotion and encouragement of IWB among employees.

### 2.4. Inclusive Leadership and Innovative Work Behavior

Leadership has long been seen as an important factor influencing creativity and innovation in organizations (Javed et al. 2018). By a variety of multiple ways, including serving as role model, providing resources (including time, funding, and information) and relationship support (George and Zhou 2007), shaping the climate (Mumford and Hunter 2005), leadership contribute to creativity and innovations.

Inclusive leadership particularly focuses on creating a safe environment in a diverse environment where all team members have the opportunity to be themselves (Nembhard and Edmondson 2006). They can spread a sense of belonging among team members through role model support for members throughout the team so that other members can replicate this caring and acceptance in team interactions. Inclusive leaders not only frequently ask for group-wide participation but also create decision-making sharing practices (Randel et al. 2018), they create a safe climate for the members to bring up new ideas and take actions to implement them. Not only do they demonstrate respect for individual group members, but they also create institutions that ensure justice and equality among groups (Randel et al. 2018). Inclusive leadership motivates team members to express their ideas and perspectives within the team by explicitly encouraging diverse discussion and communication in a fair environment, so employees are more willing to indulge in the innovative behavior.

By emphasizing the importance of uniqueness, inclusive leaders send a clear single that they expected members to share and exchange distinct qualities, and utilize these in their work practice. Their voice is appreciated and they are comfortable with speaking up and expressing themselves, employees generate increasing intrinsic motivation and cultivate energy to engage in the creative task (Atwater and Carmeli 2009). Inclusive leaders sincerely appreciate the input of employees, encourage diverse contributions and help group members fully contribute, which are described to be positively affecting the search for innovative ideas (De Jong and Den Hartog 2007), thus further augmenting the suitability of inclusive leadership for promoting IWB among the employees. Moreover, inclusive leadership due to its emphasis on both belongingness and uniqueness can cultivate a work group identification (Randel et al. 2018) thus convincing the employees to view the development and interest of the organization as their own and work for their attainment.

IWB is complex and multistage work behavior in which the implementation of ideas is an important part of innovation prosperity (Scott and Bruce 1994). If the sociopolitical process is facilitated, it has more chances of success. Inclusive leadership helps members meet their needs and express support for them and their opinions, which will lead to team members sharing information and making joint decisions about the next steps, ensuring truly shared decision-making participation within and across tasks (Randel et al. 2018). When employees are supported and connected among their colleagues and across the organizational hierarchy, employees with innovative ideas can easily reach out to others to get the support they need for their implementation, which gives them the confidence to go down the path of innovation, regardless of the risk of failure. Based on the leader-member exchange theory, by maintaining a good relationship with their leaders, employees receive work-related support in terms of social and political resources such as time and innovation-related information (Piansoongnern 2016). At the same time, with a quality relationship with their leaders, employees also experience positive feelings which triggers them to implement innovative tasks(Lin and Liu 2012). In the presence of the discussed evidence, the following hypothesis can be proposed.

**H1.** *Inclusive leadership is positively related to innovative work behavior.*

*2.5. Inclusive Leadership and Job Crafting*

High-quality leader-member exchange relationships, supporting for employee actions or decisions, providing information, consulting employees, and trusting in leaders (Atwater and Carmeli 2009) will make employees willing to seek resources to initiate and implement change processes and challenge the status quo so as to enhance their self-image. According to self-enhancement theory, the desire of individuals to maintain or increase their self-concept is seen as a basic human need (Leary 2007). In line with this, Wrzesniewski and Dutton argue that employees are encouraged to engage in job crafting behaviors with the expectation to establish and maintain a positive self-image. In line with members' needs for a possible self-image for a positive future, by demonstrating a clear focus on emphasizing participation in decision-making and facilitating diverse contributions, helping members fully contribute (Randel et al. 2018), inclusive leaders' core behaviors may increase the likelihood that their followers will proactively engage in increasing structural job resources (e.g., by actively searching for the work related information) to maintain or enhance their self-image (Parker et al. 2010).

A work setting characterized by cultivating belongingness through being accessible and available, ensuring justice and equity among members, and encouraging shared decision-making, inclusive leadership creates social bonds between leaders with members, and members with members, which could be beneficial to crafting the social dimension of one's resources. Inclusive leaders consistently engage in showing concern about employees' expectations (Wang et al. 2019), showing emotional support to the employee (Fang et al. 2021), and valuing employee's uniqueness (Randel et al. 2018). Moreover, research shows that inclusive leaders help employees cultivate a work group identification, likely to

positively influence their relationships with their group members (Randel et al. 2018). Thus, we argue that inclusive leader pays attention to his/her group members, the followers are more likely to increase their social resources (e.g., by asking their leader or other members for suggestions).

Psychologically powerful people feel they have influence and control over their actions (Spreitzer 1995). Inclusive leaders show team members that their uniqueness is respected, their perspectives are welcomed and valued and their contributions are appreciated thus providing a sense that team members have influence and control. Research has shown that being affected in the workplace enhances people's perceptions of competence and control (Boudrias et al. 2014). Inclusive leadership behavior satisfies the three psychological needs of employees, namely, competence, relationship and autonomy which are proposed by self-determination theory, so employees will be encouraged by intrinsic motivation (Deci and Ryan 2008). We therefore assume that when inclusive leaders listen to employees, promotes opportunity to share ideas on how to perform work, collaborate during decision-making with the employee, and help employees to contribute, group members are more likely to craft their jobs in the form of increasing their challenging job demands, for example by taking on new tasks. Thus, we hypothesize:

**H2.** *Inclusive leadership is positively related to employees' job crafting in the form of (a) increasing structural job resources; (b) increasing social job resources; (c) increasing challenging job demand.*

### 2.6. Job Crafting and Innovative Work Behavior

IWB includes idea creation and idea implementation. In the first stage of innovative work behavior, employees are addicted to cognitive processes. However, innovative work behavior in the second stage requires employees to be immersed in the sociopolitical process.

Occupational stress resulting from high job demands is related to physical and psychological health (Jackson and Rothmann 2006). Besides, employees with more resources will experience less stress than those with fewer resources (Salanova et al. 2010). By increasing both job resources and challenging job demands, job crafting germinates positive emotions (Costantini and Sartori 2018), which augment the thought-action repertoire of the employee. According to the broaden-and-build (B&B) theory, the broadened thought-action repertoire facilitates cognitive processes (Fredrickson 2001). More specifically, since the presence of positive emotions is associated with an expanded repertoire of thought and action, employees are expected to be more creative in the cognitive process (Fredrickson 2004).

Positive emotions and resources are proven to be effective to facilitate the sociopolitical processes, which are essential for idea implementation. Firstly, the broaden-and-build (B&B) theory indicates that positive emotions enhance the thought-action repertoire, with which, employees can better present and sell their ideas (Fredrickson 2001). In addition, according to the theory of conservation of resources (COR), resources can be invested to create other resources (Hobfoll 2002). Employees who indulge in job crafting behavior increase both structural job resources and social job resources. Building relationships with others is also a resource to facilitate sociopolitical processes (Naghavi and Mubarak 2019). To summarize, since the resulting germinates positive emotions and enables the employees to conserve resources, job crafting behavior is proposed to facilitate the sociopolitical process, thus finally help motivate and facilitate innovative work behavior. The following hypothesis is proposed.

**H3.** *Job crafting in the form of (a) increasing structural job resources; (b) increasing social job resources; (c) increasing challenging job demand is positively related to innovative work behavior.*

### 2.7. Mediating Role of Job Crafting and Hypotheses Development

By emphasizing both uniqueness and belonging, inclusive leadership has a positive influence on employees' job crafting, which results in employees generating positive emotions and increased resources so as to motivate them to indulge in innovative work behavior.

As discussed above, inclusive leaders encourage the members to pursue self-image, create a social bond within work group, and help employees feel intrinsically motivated. These behaviors subsequently enable them to pursue job crafting behavior. When employees craft their jobs, they seek to increase their job resources and job challenges (Tims and Bakker 2010). As the employees can create and mobilize resources through bottom-up approach by asking for feedback and support from supervisors, they find their jobs more enjoyable, thus increasing the likelihood of indulging in extra-role behavior (Demerouti et al. 2015). According to the theory of conservation of resources and broaden and build theory (Fredrickson 2001), employees will be more likely to indulge in innovative job behavior with positive emotions and increased resources. We rely on leader–member exchange theory to explain the mediating role of job crafting between IL on IWB. Our prediction is based on the principle that higher-quality leader-follower relationships produce more positive outcomes (Costigan et al. 2006). In addition, in high-quality relationships with leaders, employees are highly motivated to commit to extra-role behaviors that generate, promote, and implement new ideas (Volmer et al. 2012). The question is whether these two lines of influence, from inclusive leadership to positive outcomes and from job crafting to innovative work behavior, are somehow linked. Though job crafting is found to mediate the relationship between other leadership, such as servant leadership (Khan et al. 2020) and transformational leadership (Afsar and Umrani 2019), and innovative work behavior, there is no evidence delineating the mediating role of job crafting between inclusive leadership and innovative work behavior. The following hypothesis is proposed.

**H4.** *Job crafting in the form of (a) increasing structural job resources; (b) increasing social job resources; (c) increasing challenging job demand positively mediates the relationship between inclusive leadership and innovative work behavior.*

The overall model of this study is shown in Figure 1.

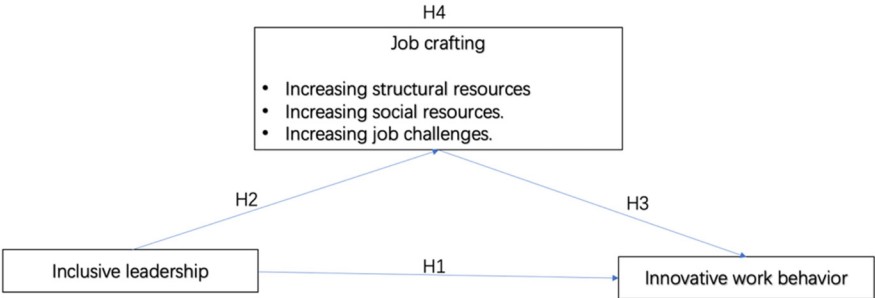

**Figure 1.** Conceptual framework.

### 3. Methods

#### 3.1. Participants and Procedure

The data used in this study were collected online, began in September 2022, and ended over a three-week period, from employees working in SMEs in China as part of a larger research project. These companies are selected because they face high pressure from complex and changing economic environment, and they are highly focused on innovation (Hajar et al. 2019). Thus, employees working in these companies have innovative jobs, and they are expected to show IWB (Afsar et al. 2017). In the innovation storm brought by the rapid development of information technology and globalization, small and medium-sized enterprises have more urgent needs for innovation, otherwise they will be eliminated by the tide of The Times. Compared with large enterprises, small and medium-sized enterprises are more flexible in their transformation, therefore, it is easier for employees to carry out innovative work behaviors and the results of innovation are also more likely to become reality. In addition, China is a country with a long history of inclusive spirit, and inclusive

culture will deepen our understanding of inclusive leadership theory and look at it from a different Angle.

A total of 358 points were collected. The final sample was 314, and 44 responses were discarded due to missing data or less than a 2-min answering time. The overall response rate was 87.7%. Since the unavailable public registry of employees, we had to use convenience sampling, which may lead to a common method bias (Vandekerkhof et al. 2019), we tried to collect data from a large and divergent sample to minimize the risk of sample bias. We collected cross-sectional data, with a final sample of 314 employees, common methodology bias may exist. We used the Harman univariate test, grouping all items into one factor for a common method bias test (Podsakoff et al. 2003). The results of the analysis showed that the total variance of the factor calculation accounted for no more than 36.7%, less than 50% (Lindell and Whitney 2001), indicating a low probability of common method bias. Spss 26.0 statistical software was used to carry out the demographic research. Among the respondents, most were employees from service companies (59%), followed by trade companies (18%), manufacturing companies (14%), and others (9%). Table 1 contains the respondents' profiles. Among all the participants, 69.4% were under 30 years old, with 72.4% less than 5 years work experience and tenure with the company. There were more female employees (73.4%) and more highly educated employees (77.6%). Table 1 shows the demographic characteristics of employees.

**Table 1.** Profile of the respondents.

| Variables | Values | Percent | Cumulative Percent |
|---|---|---|---|
| GENDER | Male | 26.6 | 26.6 |
| | Female | 73.4 | 100.0 |
| AGE | Younger than 25 | 28.0 | 28.0 |
| | 26~30 | 41.4 | 69.4 |
| | 31~35 | 9.9 | 79.3 |
| | 36~40 | 12.5 | 91.8 |
| | Older than 41 | 8.2 | 100.0 |
| EDUCATION | Undergraduate | 22.4 | 22.4 |
| | Bachelor | 60.2 | 82.6 |
| | Master | 15.5 | 98.0 |
| | PhD | 2.0 | 100.0 |
| POSITION | Ordinary employees | 56.6 | 56.6 |
| | First-line managers | 24.7 | 81.3 |
| | Middle managers | 11.2 | 92.4 |
| | Top managers | 7.6 | 100.0 |
| TENURE | 0~1 | 24.3 | 24.3 |
| | 2~3 | 30.9 | 55.3 |
| | 4~5 | 17.1 | 72.4 |
| | 6~10 | 14.1 | 86.5 |
| | 10 and more | 13.5 | 100.0 |

*3.2. Instrument Design and Measurement*

The questionnaire was distributed to employees in online format. This allowed more employees to be included. Since we adopted the original English construct, we used a back-to-back translation method into the local language to ensure the validity and accuracy of the content. The participants responded to the items using a 7-point Likert scale ranging from 1 (strongly disagree) to 7 (strongly agree).

3.2.1. Innovative Work Behavior

The six items of the innovation work behavior questionnaire developed by Scott and Bruce (1994) were used in this study. The items evaluated employees' capabilities in innovative idea generation and implementation. A sample of the items is "I will actively make plans to implement my ideas." The analysis revealed the high reliability of the items to measure the employees' innovative work behavior (CR = 0.877).

### 3.2.2. Inclusive Leadership

Inclusive leadership adopts the Inclusive Leadership Scale (Ashikali 2019), which has 13 items. Typical items include " My leaders encourage me to discuss diverse viewpoints and perspectives to problem-solving with colleagues". Aggregated measures were used to estimate team members' perceptions of leadership behavior rather than using leaders' own estimates of their intended leadership behavior. This was because previous research has found discrepancies between leader and employee ratings of leadership, it is suggested to use aggregated employee rating (Jacobsen and Bøgh Andersen 2015). (CR = 0.884).

### 3.2.3. Job Crafting

To measure job crafting behavior, a 13-item scale proposed by Tims et al. (2012) was used to assess each behavior (increasing structural job resources (JCA), increasing social job resources (JCB), and increasing challenging job demands (JCC)) (Tims et al. 2012). Typical items of increasing structural job resources "I try to learn new things at work", social job resources, "I ask for feedback on my job performance" and challenging job demands "When there is not much to do at work, I see it as an opportunity to begin new projects". The analysis revealed the high reliability of the items to measure the employees' job crafting behavior, with α reliability of increasing structural job resource (CR = 0.854), increasing social job resource (CR = 0.863), and challenging job demand (CR = 0.872), respectively.

## 4. Results

### 4.1. Measurement Model

Prior to the test of the measurement model, a confirmatory factor analysis (CFA) was performed to ensure discriminant validity of the items of each construct in this research (innovation work behavior, inclusive leadership and job crafting). Using Mplus 8, this study found that all of the items significantly loaded to the related constructs (>0.60). As Table 2 shows, all of the factor loadings are higher than 0.6, CRs are greater than 0.80 and AVEs are higher than 0.50 (Fornell and Larcker 1981). Furthermore, the square root of the AVEs is higher than their correlations with the other variables (Table 2). Therefore, the items of each construct have higher loading to the related construct than to other constructs.

**Table 2.** Reliability and Validity.

| Dim | Item Reliability | Composite Reliability | Convergence Validity | Discriminate Validity | | | | |
|---|---|---|---|---|---|---|---|---|
| | STD.LOADING | CR | AVE | IL | IWB | JCA | JCC | JCD |
| IL | 0.602~0.789 | 0.884 | 0.523 | **0.723** | | | | |
| IWB | 0.612~0.775 | 0.877 | 0.544 | 0.651 | **0.738** | | | |
| JCA | 0.726~0.828 | 0.854 | 0.594 | 0.548 | 0.680 | **0.771** | | |
| JCB | 0.674~0.797 | 0.836 | 0.506 | 0.645 | 0.310 | 0.354 | **0.711** | |
| JCC | 0.698~0.867 | 0.872 | 0.631 | 0.625 | 0.708 | 0.343 | 0.403 | **0.794** |

Note: The diagonal bold is the square root of AVE, and the lower triangle is a unique Pearson correlation. JCA refers to job crafting dimension-increasing structural job resources; JCB refers to job crafting dimension-increasing social job resource; JCC refers to job crafting dimension-increasing challenge job demand.

### 4.2. The Structural Model

To test the hypothesized relationships among the constructs in this study, a structural model was developed. Results of this model are displayed in Figure 2 and Table 3. Hypothesis 1 proposed the association between inclusive leadership and employees' innovative work behavior. Analysis of the data revealed the positive effect of inclusive leadership on employees' innovative work behavior (t = 3.167, $p < 0.005$). It was also found that inclusive leadership significantly enhances employees' increasing structural resources, as proposed by H2 (1) (t = 11.185, $p < 0.001$). The analysis also supported H2(2) that hypothesized inclusive leadership significantly improves the behavior of increasing social resources (t = 14.851, $p < 0.001$). In addition, this study confirmed the hypothesized H2(3), which

posits the positive effect of inclusive leadership on increasing challenging job demands (t = 14.560, *p* < 0.01). H3 proposed the significant influence of increasing structural resources[H3(1)], increasing social resources [H3(2)], and challenging job demand[H3(3)] on employees' innovative work behavior was also confirmed by the data (t = 8.433, *p* < 0.001), (t = −2.873, *p* < 0.005)and (t = 8.247, *p* < 0.001)respectively. However, it is worth noting that increasing social job resources has a negative effect on innovative work behavior, which is not consistent with our hypothesis. Therefore, H2 was also supported and H3 was partially supported.

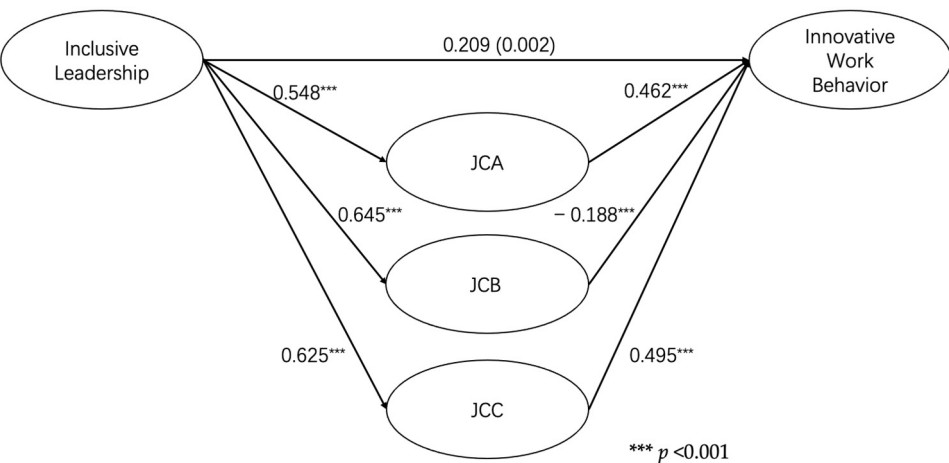

**Figure 2.** Path diagram.

**Table 3.** Structural model.

| DV | IV | Estimate | S.E. | Est./S.E. | *p*-Value | R-Square | HYPO |
|-----|------|----------|-------|-----------|-----------|----------|------|
| JCA | IL | 0.548 | 0.049 | 11.185 | *** | 0.300 | S |
| JCB | IL | 0.645 | 0.043 | 14.851 | *** | 0.416 | S |
| JCC | IL | 0.625 | 0.043 | 14.560 | *** | 0.391 | S |
| IWB | IL | 0.209 | 0.066 | 3.167 | 0.002 | 0.743 | S |
| | JCA | 0.462 | 0.055 | 8.433 | *** | | S |
| | JCB | −0.188 | 0.065 | −2.873 | 0.004 | | S |
| | JCC | 0.495 | 0.060 | 8.247 | *** | | S |

Note: *** *p* < 0.001; JCA refers to job crafting dimension-increasing structural job resources; JCB refers to job crafting dimension-increasing social job resource; JCC refers to job crafting dimension-increasing challenge job demand.

### 4.3. The Mediating Effect

Finally, H4 proposes that job crafting behaviors (increasing structural job resources, increasing social job resources and increasing challenging job demands) mediate the relationship between inclusive leadership and employees' innovative work behavior. In order to test this effect, we should compare the total effect of inclusive leadership on employees' innovative work behavior and the indirect effect between them.

As shown in Table 4, as for the total effect of inclusive leadership on employees' innovative work behavior, the 95% confidence interval of bootstrap 5000 results did not include 0, it means that there is a direct relationship between inclusive leadership and innovative work behavior. The indirect influence of inclusive leadership on innovative work behavior through the mediating effect of different job crafting behaviors is statistically significant and the 95% confidence interval of bootstrap 5000 results did not include 0. Therefore, our mediation hypothesis that job crafting in the form of (a) increasing structural job resources; (b) increasing social job resources; (c) increasing challenging job demands mediates the relationship between inclusive leadership and innovative work behavior is therefore supported by considering the two conditions established by Preacher and Hayes

(2004). However, the dimension of increasing social job resources is a negative mediatory, which is inconsistent with our hypothesis.

**Table 4.** The effect of inclusive leadership on IWB through job crafting behaviors.

| | Pint Estimate | Product of Coefficients | | Bootstrap 5000 Times 95% | |
| | | | | Bias Corrected | |
| | | SE | Est./S.E. | Lower | Upper |
| --- | --- | --- | --- | --- | --- |
| | | INDIRECT EFFECT | | | |
| JCA | 0.282 | 0.071 | 3.96 | 0.164 | 0.448 |
| JCB | −0.135 | 0.08 | −1.69 | −0.342 | −0.002 |
| JCC | 0.345 | 0.088 | 3.931 | 0.213 | 0.557 |
| TIE | 0.492 | 0.108 | 4.558 | 0.325 | 0.746 |
| | | TOATL EFFECT & DERECT EFFECT | | | |
| TE | 0.725 | 0.093 | 7.806 | 0.562 | 0.93 |
| DE | 0.233 | 0.109 | 2.15 | 0.016 | 0.44 |

Note: JCA refers to job crafting dimension-increasing structural job resources; JCB refers to job crafting dimension-increasing social job resource; JCC refers to job crafting dimension-increasing challenge job demand.

## 5. Discussion and Conclusions

In this study, we developed and tested a model in which inclusive leadership and job crafting in the form of (a) increasing structural job resources (b) increasing social job resources and (c) increasing challenging job demands were hypothesized to affect innovative work behavior directly and indirectly. The first was to illuminate the role of inclusive leadership in promoting employee innovative work behavior. The second was to illustrate the relationship between inclusive leadership and job crafting. Third, the study is intended to ascertain the positive effect of job crafting on innovative work behavior. Fourth, the study aimed to test whether job crafting mediates the relationship between inclusive leadership and innovative work behavior.

First of all, the findings confirm the claim that inclusive leadership is related to innovative work behavior. IWB is an unconventional high-risk performance in which employees avoid traditional work activities and express their opinions on new work technologies. Therefore, employees need to establish a quality relationship with leaders in order to challenge existing standard operating procedures (Janssen 2005). In the context of innovation, inclusive leaders share important knowledge and help employees generate, recognize and execute useful ideas (Yeoh and Mahmood 2013). When inclusive leaders also focus on morality by encouraging employee participation in decision-making, employees feel more trustworthy (Hollander 2013). Thus, a quality LMX including mutual support, emotional connection, and loyalty (Wayne et al. 2002) explained the principle behind it, and as a result, employees show innovative work behavior. Inclusive leadership, through its emphasis on belonging and uniqueness, helps to create a safe environment for employees to voice their opinions and participate freely, and to contribute fully to engaging employees in innovative work behaviors. The finding of the study was in line with the findings of earlier studies that related inclusive leadership with innovative work behavior (Javed et al. 2018, Javed et al. 2019c). Second, the study found evidence for the relationship between inclusive leadership and job crafting behavior (decreasing hindering job demand is not included, the following is the same). Self-determination theory states that people must constantly meet three basic psychological needs–autonomy, competence and relatedness–throughout their lives in order to achieve optimal levels of functioning and to experience sustained personal growth and well-being (Vallerand 2000). Inclusive leadership promotes a sense of belonging by involving employees in decision-making and fostering work group identification and promotes an emphasis on uniqueness by helping team members contribute fully, thus meeting basic human needs. Humans' pursuit of life goals driven by intrinsic desires can improve well-being by fulfilling basic needs.(Ryan et al. 1996). Employees who meet the three basic needs are intrinsically motivated and tend to pursue extra-role behaviors

thus implementing job crating. Third, the study found job crafting to be related to the innovative work behavior of the employees. The result of the study partially corroborated the earlier findings depicting a relationship between job crafting and innovative work behavior (Khan et al. 2020). In our study, the job crafting behavior of increasing structural job resources and increasing challenging job demands are positively related to innovative work behavior. However, increasing social job resources is negatively related to innovative work behavior. This can be interpreted as an over-reliance on social resources leading to behavioral and intellectual laziness among employees, thus hindering employees' work innovation behavior.

In addition to direct effects, the study also attempts to test the mediating role played by job crafting in the first place. Job crafting triggered by inclusive leadership injects positive emotions into employees and increases employees' available resources. Employees who take advantage of positive emotions and increased structural job resources are more likely to indulge in innovative work behaviors. However, over-reliance on social job resources will negatively mediate the relationship between inclusive leadership and innovative work behavior. While job crafting has not been mentioned as a mediator between inclusive leadership and innovative work behavior, its role as a mediator between inclusive leadership and pro-organizational behavior has been well documented (Bavik et al. 2017). The current study's finding adds to the existing evidence for a mediating role of job crafting linking inclusive leadership and pro-organizational work behavior.

Despite somewhat conflicting results, our study makes an important contribution in two ways. First, it lends empirical support to managerial theories of the motivational effects of inclusive leadership with respect to follower job crafting and innovative work behavior. Second, we learned that (1) job crafting behavior can be triggered by inclusive leadership and mediated the effect of inclusive leadership on innovative work behavior. (2) over-rely on social resources may lead to laziness and have a negative effect on innovative work behavior. (3) Since job crafting–increasing social work resources is negatively correlated with innovative work behavior, the mediating effect is also negatively correlated. That is to say, the more inclusive the leader's behavior, the more likely the employee is to increase their social resources, and the more over-reliance on social resources, the less inclined the employee is to indulge in innovative work behaviors.

Although this study is only conducted in China, with globalization, countries around the world continue to exchange culture and management experience and gradually converge. Therefore, in addition to guiding management practices in China, this study is also of practical significance to countries around the world, especially those with collectivist cultural backgrounds.

### 5.1. Theoretical Contribution

The present study adds to existing theory mainly in three ways. First, as more and more evidence emerges on the relationship between inclusive leadership and innovative work behavior, the need to understand the mechanism of this link becomes increasingly clear. The mediation path test in this study was conducted by job crafting has not been studied before.

Although job crafting is considered a bottom-up approach, our findings suggest that similar to other proactive behaviors, leaders can actually stimulate job planning by demonstrating inclusive leadership. Through its inclusion philosophy, inclusive leaderships adopt practices such as supporting individuals as group members, ensuring justice and equity, and sharing decision-making to help members generate social bonds within the group to increase social resources, at the same time, inclusive leaders also concentrate on encouraging diverse contributions and helping members fully contribute to enhance group members' self-image and meet their need for competence, so encourage members to increase both structural resources and job challenges. The positive result of job crafting is that members gain positive emotions and increased resources, which also provides a necessary prerequisite for employees to indulge in innovative behavior.

Second, our operationalization of job crafting is quite different from previous studies. We studied its behavioral dimensions separately instead of seeing job crafting as a high-ordered construct. This allows us to assess the influence of various aspects of a particular construct, to understand in detail the influence of each of the different dimensions, and to enrich the understanding of the job crafting theory.

Third, we test our suggested framework in China. One of the important reasons is its inclusive spirit which enables its culture passed on for thousands of years. Studying inclusive leadership in fertile soil with thousands of years of inclusive culture will undoubtedly provide richer connotations and different aspects to this theory. It will also further deepen our understanding of innovative work behavior and job crafting in the context of the unique cultures that Asian countries present.

Additionally, the study adds to the existing empirical evidence related to the relationship between inclusive leadership and innovative work behavior. So far, research on the impact of leadership on creativity is still in its infancy (Tierney 2008). Our research empirically confirms the positive impact of inclusive leadership on innovative work behavior, and through our research, we reshape and consider this mechanism through job crafting as a mediating variable, thus providing a meaningful supplement to this theory.

### 5.2. Practical Contributions

This study has multiple practical implications. First, this study confirms the findings of earlier studies that inclusive leadership promotes innovative work behavior. Inclusive leadership provides a safe climate and adequate support for members' innovative behavior by emphasizing both belongingness and uniqueness. Small and medium-sized enterprises face more intense challenges in their development, but their advantage is that the transformation is more flexible. Therefore, leaders of small and medium-sized enterprises should adopt an inclusive approach to encourage employees to actively participate in innovative work behavior and help their enterprises stand out in the ever-changing market environment.

Second, this study proves that job crafting plays a mediating role in the influence of inclusive leadership on innovative behavior. Therefore, in practice, inclusive leadership can motivate and guide employees to actively participate in job crafting behaviors through purposeful incentives, such as adopting a more open attitude, providing necessary resources for employees to craft their work behavior, and helping create an equitable atmosphere of mutual assistance. Leaders also should recognize the diverse contributions of employees and provide the necessary support for their job crafting behavior.

Research has also found that job crafting is a double-edged sword (Kim et al. 2018). In order to maintain a good result for job crafting, organizations should focus on hiring highly motivated employees with flexibility in skills and behaviors to adapt to new roles and be more willing to make positive changes in the workplace. Employees should also be trained to master a broad set of skills and be able to use them in different demand conditions. At the same time, inclusive leadership should pay attention to the reasonable division of labor to avoid employee over-reliance on social job resources, which may lead to evasion and irresponsibility, and ultimately affect employees' innovative behaviors. Our findings suggest that inclusive leadership leverages the positive aspects of job crafting and limits the negative aspects.

### 5.3. Limitations and Future Studies

One possible limitation of our study is that it was conducted in China, a country with a long history of inclusiveness. It is well-established that culture affects workplace processes, including leadership (Rockstuhl et al. 2012). The need for belonging and uniqueness is thought to be universal (Fromkin and Snyder 1980), but in different cultural contexts, ways of meeting these needs may vary. Future research needs to examine the role and function of inclusive leadership in various cultural contexts, while reflecting cultural elements considered socially appropriate.

Second, The current study collected all the data at once, which could have led to a common method bias (MacKenzie and Podsakoff 2012). We assessed the effects of common method bias using confirmatory factor analyses. This set of analyses may provide some indication that in our study, the common method variance is not a severe problem, but future research is still recommended to apply a longitudinal study and use different sources of data, collect data at least at two points in time to avoid the common method bias.

Third, despite our focus on inclusive leadership and job crafting, we recognize that other unseen variables may be critical in explaining employees' innovative work behavior. Therefore, future research may integrate complementary theories and explanations of innovative work behavior. For example, cognitive abilities and job characteristics may promote innovative work behavior. In addition, leader inclusion may affect the sense self-efficacy and goal self-concordance may also affect innovative work behavior. It is therefore important to look for a more integrated approach to understanding how inclusive promotes innovative work behavior.

**Author Contributions:** Conceptualization, Y.G.; methodology, Y.G.; software, Y.G.; validation, Y.G., J.J. and S.-H.Y.; formal analysis, Y.G.; investigation, Y.G.; resources, J.J.; data curation, Y.G.; writing—original draft preparation, Y.G.; writing—review and editing, Y.G.; visualization, J.J.; supervision, S.-H.Y.; project administration, S.-H.Y.; funding acquisition, Y.G.. All authors have read and agreed to the published version of the manuscript.

**Funding:** This research received no external funding.

**Institutional Review Board Statement:** Not applicable.

**Informed Consent Statement:** Informed consent was obtained from all subjects involved in the study.

**Data Availability Statement:** The data presented in this study are available on request from the corresponding author. The data are not publicly available due to copywrite problem.

**Conflicts of Interest:** The authors declare no conflict of interest.

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
