# Peer review of "Impact of Inclusive Leadership on Innovative Work Behavior: The Mediating Role of Job Crafting"

_admsci, doi:10.3390/admsci13010004_

Round 1

Reviewer 1 Report

The study examines the relations between includive leadership, job craftin and innovative work behavior (IWB). In addition, the study tests the mediation of job crafting in the relation between includive leadership and IWB. The article is well structured, the contribution of the study is clear and the methodology is well described and appropriate. I found the article very intersting.

Author Response

Dear Reviewer:

Thank you very much for your appreciation and encouragement. I will make continuous efforts in future research work.

Best regards
YINPING GUO

Reviewer 2 Report

The paper is interesting, well-written, and has scientific validity. The author defined the goal, methodology, presented the results, and conclusions, and presented the contribution and limitations and further research directions.

I only would suggest some points to make the paper better:

1)    The theoretical background needs to add to explain the relationship between the research constructs

2)    Please address in 2.2, why you use 3 dimensions for Job crafting

Author Response

Dear reviewer:

Thank you for your decision and constructive comments on my manuscript. We have carefully considered the suggestions and make some changes. Each of your suggestions has prompted me to study more extensively and think more deeply. I cherish your efforts very much. 

Reviewer 3 Report

Thank you for submitting the manuscript id admsci-2060678 entitled “Impact of inclusive leadership on innovative work behavior: The mediating role of job crafting.” A plethora of research is available on these variables e.g., inclusive leadership and IWB. Thus require a substantial gap to get the interest of the researchers and practitioners.

Good luck

Introduction

The gap needs to be identified properly. As per the author's argument, “studies assessing inclusive leadership and  IWB are scarce and generally do not analyze the mechanisms by which inclusive leadership may influence employee creative engagement.” Actually, it is not the case, there are many studies available that checked the mediating mechanism between inclusive leadership and  IWB.

E.g.,

Bannay, D. F., Hadi, M. J., & Amanah, A. A. (2020). The impact of inclusive leadership behaviors on innovative workplace behavior with an emphasis on the mediating role of work engagement. Problems and Perspectives in Management18(3), 479.

Qi, L., Liu, B., Wei, X., & Hu, Y. (2019). Impact of inclusive leadership on employee innovative behavior: Perceived organizational support as a mediator. PloS one14(2), e0212091.

Fang, Y. C., Chen, J. Y., Wang, M. J., & Chen, C. Y. (2019). The impact of inclusive leadership on employees’ innovative behaviors: the mediation of psychological capital. Frontiers in Psychology10, 1803.

Javed, B., Abdullah, I., Zaffar, M. A., ul Haque, A., & Rubab, U. (2019). Inclusive leadership and innovative work behavior: The role of psychological empowerment. Journal of Management & Organization25(4), 554-571.

Javed, B., Khan, A. K., & Quratulain, S. (2018). Inclusive leadership and innovative work behavior: examination of LMX perspective in small capitalized textile firms. The Journal of Psychology152(8), 594-612.

Javed, B., Naqvi, S. M. M. R., Khan, A. K., Arjoon, S., & Tayyeb, H. H. (2019). Impact of inclusive leadership on innovative work behavior: The role of psychological safety. Journal of Management & Organization25(1), 117-136.

Therefore, the gap is not significant and better arguments are required.

 Literature review

In general, this section is well written. However, I believe the arguments leading to mediating hypothesis are weak and need to be strengthened. In addition, I suggest the authors add recent literature to make it strong. Furthermore, use LMX theory in developing the hypotheses

Methods and results

  • Regarding methods, these things need to be elaborated

Could you please justify why the SMEs in China are targeted?

Which method was used for calculating the sample size?

When the data was collected?

Since the data were cross-sectional, how did you avoid common method bias (CMB) during data collection?

Why MPlus and why not CB-SEM and PLS-SEM?

Please make explicit the concrete measures you use. We cannot and should not guess what variables you are using for your Mplus, and it is not satisfying that you hide them for the reader behind the references. Show them e.g. in a table. This will also enable you to show us the SEM model and potential path diagrams related to it. What I am asking for here is transparency and the explicit presentation of concrete variables/measures for the constructs you use. The study would stand much stronger with this information available.

  • Results are fine

Discussion and implication

Please explain your results with the help of the adopted theory (e.g., LMX theory).

What specific conclusion is drawn from your study and how your study would be fruitful in
the international context?

Theoretical and practical implications are well explained.

Other comments

Please check the in-text citation for typos

Author Response

(The authors gave the same response as above.)

Round 2

Reviewer 3 Report

Thank you for submitting the revised manuscript id admsci-2060678 entitled “Impact of inclusive leadership on innovative work behavior: The mediating role of job crafting.” I appreciate the author(s) effort in revising the manuscript, however, there is still room for improvement. See Pdf
